# LEARNING FAIR LATENT REPRESENTATION WITH MULTI-TASK DEEP LEARNING

## ABSTRACT

The problem of group-level fairness in machine learning has received increasing attention due to its critical role in ensuring the reliability and trustworthiness of models deployed in sensitive domains. Mainstream approaches typically incorporate fairness by enforcing constraints directly within the training objective. However, treating fairness solely as a regularisation term can lead to suboptimal trade-offs with loss of accuracy or insufficient fairness guarantees. In this work, we propose a novel approach that formulates fairness as an auxiliary task in a Multi-Task Learning (MTL) paradigm. In contrast to embedding fairness constraints into a single-task objective, explicitly modelling the problem as multi-objective optimisation (MOO) allows to decouple the learning of a fair internal representation from the optimisation of the predictive task: these two conflicting objectives are optimised concurrently. We introduce two novel fairness loss functions that are better tailored to an MTL approach. We provide a theoretical analysis of the generalisation properties of the proposed approach. The experimental analysis on benchmark datasets shows that in spite of not embedding a fairness loss function directly on the predictive task the MTL formulation consistently improves group-level fairness metrics compared to both standard regularisation-based methods and other MTL architectures, while maintaining competitive predictive performance. Code is available at https://anonymous.4open.science/r/EDK1-02EB.

## 1 INTRODUCTION

With the widespread integration of machine learning (ML) into decision-support systems, ensuring equitable and unbiased outcomes is vital for maintaining system integrity and trust. Historical data biases often seep into ML models, perpetuating inequities in evaluations and applications. To counter this, fairness-driven learning strategies have been developed in three main categories. Pre-processing approaches attempt to reduce bias before training by modifying the data (e.g., sampling, weighting, feature transformation). These are useful when the data is the main source of bias. However, the potential loss of information in this approach can limit the learnable features in large datasets. Post-processing methods are useful when the model is already trained and cannot be easily retrained, although altering model outputs to reduce bias may affect model performance as well as interpretability. In-processing approaches incorporate fairness constraints into the learning algorithm, e.g., by integrating some fairness measure into the training objective. A typical approach introduces a fairness regularisation term in the loss function. Other techniques are based on adversarial debiasing or, more generally, to some form of fairness-aware optimisation. These methods can usually achieve the strongest and most principled fairness guarantees since constraints are built into the optimisation process. Most importantly, this approach introduces an explicit trade-off between fairness and accuracy of the predictive task.

In this work, we investigate how to enhance fairness in the internal/latent representation and the effectiveness of an in-training approach where the predictive component of the model is not directly subject to fairness optimisation or constraints.

Regularisation and constraint optimisation are significant in-processing techniques. Regularisation embeds fairness penalties in the objective function for adaptable fairness alignment, though it may affect robustness Zemel et al. (2013). Constraint optimisation enforces fairness as a strict constraint but can be destabilised by conflicting constraints Cotter et al. (2019). An alternative approach is

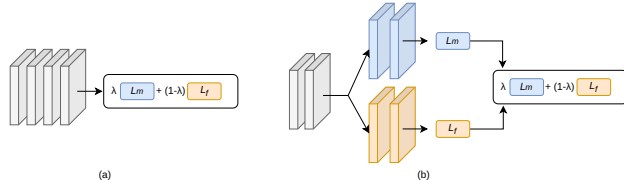

Figure 1: Illustration of the two learning paradigms with the main task loss denoted as $\mathcal{L}_m$, the fairness task loss denoted as $\mathcal{L}_f$, and the fairness regularisation hyperparameter $\lambda$. (a) *STL learning*: Group-level fairness learned via regularisation optimisation, $\mathcal{L}_f$ is not mandatory to be a loss function and can also be a penalty score. (b) *MTL learning*: Fairness is learned concurrently to the main predictive task by means of two distinct task-specific layers and a shared latent representation.

learning a latent representation that accurately encodes the target variable while remaining neutral to sensitive attributes Madras et al. (2018); Adel et al. (2019). Adversarial debiasing seeks a fair representation invariant to sensitive attributes, yet it demands careful balancing of predictor and adversary objectives, increasing training complexity.

To surmount the challenges and shortcomings of previous methods, we propose framing fair latent representation as a multi-objective optimisation problem, dealing with two distinct types of tasks, primary predictive tasks and fairness tasks, through Multi-Task Learning ($MTL$) Caruana (1997). $MTL$ effectively balances objectives, avoiding penalties and constraints of Single-Task Learning ($STL$).

Unlike methods adopting fairness regularisation in a single objective function, we explore fairness regularisation as an auxiliary concurrent task, transitioning from $STL$ to an $MTL$ framework (see Fig. 1). During training, the two objective functions of the main prediction task and of the fairness task are combined. However, each task-specific component of the architecture is subject only to the task-specific gradient, while the shared component is optimised with their aggregation inducing a fair and effective latent representation.

The underlying rationale of the auxiliary fairness task is to induce the shared layers to learn a latent representation in which information about the sensitive variable is encoded without bias. Consequently, the main task learns from an unbiased latent representation without being directly penalised by a direct fairness objective.

Overall, this work aims to bridge the gap between fairness and performance in ML models by leveraging $MTL$ and provides the following contributions.

- **Fairness as an auxiliary task:** We incorporate group-level fairness into a multi-task learning framework as an auxiliary objective, systematically evaluating its impact on both predictive performance and fairness metrics.

- **Novel fairness loss functions:** We introduce two new fairness-oriented loss functions designed to guide model optimisation, analyse their effectiveness in promoting fair outcomes.

- **Theoretical guarantees:** We derive explicit generalisation bounds showing that optimising the multi-task objective provably reduces the true group fairness gap. Our analysis connects optimisation error, model capacity (via Rademacher complexity), and sample imbalance to fairness performance, establishing the first rigorous optimisation-to-fairness guarantees in the MTL paradigm.

- **Experimental analysis:** We compare the proposed approach to $STL$ methods adopting fairness regularisation as well as other existing $MTL$ methods for group-level fairness.

The reminder of this work is organised as follows. In section 2 we discuss related work and present a general definition of fairness and the MTL paradigm. In section 3 we introduce the concept of auxiliary fairness tasks for sensitive variables and two novel fairness loss funtions. In section 4 a theoretical analysis of the proposed approach is carried out. Sections 5 and 6 present, respectively, the setup of the experimental analysis and the results. Finally, section 7 provides some remarks and outlines future research directions.

## 2 RELATED WORK

Group-level fairness in machine learning has gained attention recently Rabonato & Berton (2024); Pessach & Shmueli (2022); Caton & Haas (2024). Advances in fairness-aware machine learning have highlighted $MTL$ as a promising method for improving group-level fairness without compromising predictive power. Research strategies for enhancing fairness through $MTL$ fall into three main categories: regularisation, optimisation, and specialised architectures to mitigate unfairness.

Regularisation strategies focus on embedding fairness constraints within $MTL$ frameworks. One method involves a $MTL$ architecture that learns both a universal model and group-specific models, directly incorporating fairness into shared components Oneto et al. (2019). Another approach uses low-rank matrix factorisation to ensure Demographic Parity (DP) in factor representation Oneto et al. (2020), while a similar technique employs Wasserstein barycenters for equitable representations in classification and regression tasks Hu et al. (2023). Optimisation strategies tackle fairness by balancing tasks and employing dynamic weighting to adjust between fairness and task-specific losses Li et al. (2023). Rank-based fairness in $MTL$ regression uses the Mann-Whitney U statistic to address bias without distributional assumptions Zhao & Chen (2019).

Task-group branching clusters tasks with similar parameters to reduce negative and bias transfer issues in fairness-aware $MTL$ Roy et al. (2024). Fairness is also explored in a multi-objective context, balancing fairness with classification accuracy Ruchte & Grabocka (2021). Recent methodologies employ bi-level optimisation to find Pareto-optimal solutions between fairness and predictive performance Yazdani-Jahromi et al. (2024).

Fairness-aware $MTL$ designs aim to mitigate bias transfer and improve fair representation. One technique trains distinct models for each group to tackle data shortages in underrepresented groups Dwork et al. (2018). Attention-aware models concurrently predict main tasks and sensitive attributes, minimising bias through attention mechanisms Majumdar et al. (2021). Another strategy leverages teacher-student models to embed fairness in the student network, balancing between primary and fairness goals Roy & Ntoutsi (2023). Adversarial debiasing methods prevent biased encodings while preserving task performance Adel et al. (2019).

Collectively, these works acknowledge that fairness can be effectively integrated into $MTL$ frameworks. Yet, most methods either impose fairness as a constraint or introduce fixed fairness objectives. Our work, however, models group-level fairness as an auxiliary task within the $MTL$ approach, offering flexible adaptation to various fairness definitions and training dynamics. Furthermore, unlike the bi–level optimization formulation Yazdani-Jahromi et al. (2024), there is no outer optimisation task and an inner optimisation task. In MOO all tasks are at the same level with no hierarchy and are optimised concurrently.

### 2.1 FAIRNESS DEFINITION

In general, a single universal definition of fairness in machine learning is neither attainable nor desirable. Instead, the choice of a fairness definition should be context-dependent, tailored to the specific application scenario Global Future Council on Human Rights 2016–18 (2018).

For the purpose of this work, we adopt a commonly used fairness definition from the literature Barocas et al. (2023); Sarhan et al. (2020); Zemel et al. (2013), which requires the predictive distribution $P(y \mid x)$ of the classifier to be statistically independent of the sensitive attribute $s$, as formalised in equation 1.

$$P(y|x) = P(y|x,s) \tag{1}$$

The primary objective is therefore to learn a fair representation that is both informative for the main prediction task and invariant with respect to the sensitive attribute.

### 2.2 MULTI-TASK LEARNING

Let $\mathcal{X} = \{(x_i, y_i^{(1)}, \ldots, y_i^{(K)})\}_{i=1}^N$ be a dataset of $N$ input samples, where each $x_i \in \mathbb{R}^d$ is associated with task-specific labels $y_i^{(k)}$ for $k = 1, \ldots, K$.

A hard-parameter sharing $MTL$ model consists of shared parameters $\theta_{Sh} : \mathbb{R}^d \to \mathbb{R}^p$, which map some input $x$ into a shared latent representation $h(x) \in \mathbb{R}^p$, and $K$ task-specific parameter sets $\theta_T^k : \mathbb{R}^p \to \mathbb{R}^{o_k}$ for each task $k = 1, \ldots, K$. Therefore the parameters of the $MTL$ model can be written as $\theta = \theta_{Sh} \cup \{\theta_T^k\}_{k=1}^K$, where $\hat{y}^{(k)} = f_{\theta_T^k}(f_{\theta_{Sh}}(x))$ transforms the shared latent representation $h(x) = f_{\theta_{Sh}}(x)$ into a task-specific output $\hat{y}^{(k)}$ for each task $k$.

The objective of training an $MTL$ model is to minimise the sum of the task-specific loss functions as defined in equation 2:

$$\arg\min_\theta \left\{ \mathcal{L}_{mtl}(\theta) := \sum_{k=1}^K \mathcal{L}_k \left( f_{\theta_T^k}(f_{\theta_{Sh}}(x)) \right) \right\}, \tag{2}$$

where $\mathcal{L}_k$ is the loss function for task $k$.

## 3 AUXILIARY FAIRNESS TASKS

In this work, we propose a hard-parameter-sharing $MTL$ architecture in which the main prediction tasks are augmented by auxiliary fairness tasks, one for each sensitive attribute, and are all optimised concurrently. The aim of the auxiliary tasks is to enhance group-level fairness in the shared representation $h(x)$, while retaining the encoding of the relevant information for the main tasks.

Let $\mathcal{X} = \{(x_i, y_i^{(1)}, \ldots, y_i^{(K)}, z_i^{(1)}, \ldots, z_i^{(S)})\}_{i=1}^N$ be a dataset of $N$ observations, where each $x_i \in \mathbb{R}^d$ is associated with $K$ task-specific labels $y_i^{(k)}$ for $k = 1, \ldots, K$, and $S$ sensitive attributes $z_i^{(s)}$ for $s = 1, \ldots, S$.

Our model architecture consists of a single set of shared parameters $\theta_{Sh} : \mathbb{R}^d \to \mathbb{R}^p$, $K$ task-specific parameter sets $\theta_T^k : \mathbb{R}^p \to \mathbb{R}^{o_k}$ and $S$ fairness-specific parameter sets $\theta_F^s : \mathbb{R}^p \to \mathbb{R}^{o_s}$ for $s = 1, \ldots, S$. It follows that the set of parameter can be written as $\theta = \theta_{Sh} \cup \{\theta_T^k\}_{k=1}^K \cup \{\theta_F^s\}_{s=1}^S$, where $f_{\theta_T^k}(f_{\theta_{Sh}}(x))$ produces the output $\hat{y}^{(k)}$ for the task $k$ and $f_{\theta_F^s}(f_{\theta_{Sh}}(x))$ produces the fairness output for the sensitive attribute $\hat{z}^{(s)}$.

The learning objective is to minimise the sum of task-specific loss functions and fairness-specific loss functions, as formalised in equation 3.

$$\min_\theta \left\{ \mathcal{L}_{\text{MTL}}(\theta) := \lambda \sum_{k=1}^K \mathcal{L}_k(\theta_T^k, \theta_{Sh}; x) + (1 - \lambda) \sum_{s=1}^S \mathcal{L}_s(\theta_F^s, \theta_{Sh}; x) \right\}, \quad \lambda \in [0, 1]. \tag{3}$$

Here, $\mathcal{L}_k$ denotes the loss for task $k$, $\mathcal{L}_s$ denotes the fairness loss associated with sensitive attribute $z^s$, and $\lambda$ is a hyperparameter controlling the trade-off between task performance and fairness. During training the task-specific parts of the architecture are not affected by the fairness loss functions, while the shared part of the model is affected by both types of objectives.

### 3.1 FAIRNESS LOSS FUNCTIONS

Fairness loss functions are commonly used in the literature to incorporate fairness into model objectives Caton & Haas (2024). These functions are typically based on fairness metrics to address disparities among sensitive groups. However, they often assume an $STL$ framework, where fairness regularisation aligns with classification predictions in the entire architecture. This approach does not fit our $MTL$ architecture, which separates prediction accuracy from fairness as objectives in distinct parts of the architecture. Using standard fairness loss functions in the MTL approach would require an additional predictive task head dedicated to fairness, leading back to optimisation issues and potentially suboptimal performance. Therefore, we introduce two new fairness loss functions that operate independently of the predictive tasks and can be integrated seamlessly within the $MTL$ framework.

The concept of a *group-specific loss* forms the basis of our approach. Given a sensitive attribute $z^{(s)}$ with groups $g \in \{1, \ldots, G_s\}$, we define the group-specific loss $\mathcal{L}_g^{(s)}$ as the task loss $\mathcal{L}_s$ restricted to the subset of samples belonging to group $g$, as shown in equation 4.

$$\mathcal{L}_g^{(s)} = \mathcal{L}_s \left( \theta_F^s \left( \theta_{Sh} \left( \mathcal{X}_g^{(s)} \right) \right) \right) \tag{4}$$

where $\mathcal{X}_g^{(s)} = \{(x, y) : z^{(s)} = g\}$. For instance, if the primary task is binary classification with a binary cross-entropy (BCE) loss and the sensitive attribute $z^{(s)}$ corresponds to gender, then $\mathcal{L}_{\text{male}}$ denotes the BCE evaluated exclusively on male samples.

**Group Loss Fairness (GLF)**  This objective computes the main task loss independently for each sensitive attribute and minimises the disparity among these group-specific losses. By directly penalising inter-group performance differences, $GLF$ promotes fairness across groups without relying on a shared classification head. The formulation is given in equation 5.

$$\mathcal{L}_{\text{fair}} = \begin{cases} \left| \mathcal{L}_{g_1}^{(s)} - \mathcal{L}_{g_2}^{(s)} \right|, & \text{if } G = 2 \\ 1 - \dfrac{\left( \sum_{i=1}^{g} \mathcal{L}_{g_i}^{(s)} \right)^2}{G \cdot \sum_{i=1}^{G} \mathcal{L}_{g_i}^{(s)^2}}, & \text{otherwise} \end{cases} \tag{5}$$

For binary sensitive attribute ($G = 2$), we use the absolute difference of the mean losses for the two groups. For multi sensitive attribute ($G > 2$), we adopt Jain's Fairness Index (JFI) Jain et al. (1984), a standard measure in resource allocation fairness, and convert it into a loss by subtracting it from 1 (i.e., $1 - \text{JFI}$).

**Group Loss Divergence (GLD)**  This objective promotes equitable treatment across sensitive groups by minimising the Kullback–Leibler divergence Kullback & Leibler (1951) ($D_{\text{KL}}$) equation 6 between the empirical distribution of group-specific losses, denoted $P$, and a uniform distribution $U$ representing ideal fairness.

$$D_{\text{KL}}(P \,\|\, U) = \sum_{g=1}^{G} P_g \log \frac{P_g}{U_g}, \quad \text{where} \quad U_g = \frac{1}{G}, \quad P_g = \frac{\exp\left( -\mathcal{L}_g^{(s)} \right)}{\sum_{i=1}^{G} \exp\left( -\mathcal{L}_i^{(s)} \right)}. \tag{6}$$

By aligning $P$ with $U$, $GLD$ encourages balanced performance across groups. The formulation is provided in equation 7.

$$\mathcal{L}_{\text{fair}} = e^{D_{\text{KL}}(P \,\|\, U)} - 1 \tag{7}$$

Here, $P$ is derived by applying a softmax transformation to negative group losses, effectively assigning higher weights to groups with smaller losses. The resulting fairness loss is then scaled exponentially, which amplifies the penalty for larger divergences while maintaining non-negativity.

## 4 THEORETICAL ANALYSIS OF MULTI-TASK FAIRNESS

We now analyse the generalisation properties of the proposed MTL objective in equation 3. Our goal is to establish provable connections between optimisation error, hypothesis class complexity, and group-level fairness. To make the exposition concrete, we focus on binary sensitive attributes ($s \in \{0, 1\}$), with extensions to multi-group settings following by standard reductions.

Let $\mathcal{H}$ denote the hypothesis class for the main classifier $f_\theta \in \mathcal{H}$ induced by parameters $\theta$. For each group $j \in \{0, 1\}$, let $P_j := P(\cdot \mid s = j)$ and $\widehat{P}_j$ be the corresponding empirical distribution over a sample $D_j = \{(x_{j,i}, y_{j,i})\}_{i=1}^{n_j} \overset{\text{i.i.d.}}{\sim} P_j$. We denote the true and empirical group losses by

$$L_j^{(s)}(f_\theta) := \mathbb{E}_{(x,y) \sim P_j} \left[ \ell(f_\theta(x), y) \right],$$

$$\widehat{L}_j^{(s)}(f_\theta) := \frac{1}{n_j} \sum_{i=1}^{n_j} \ell\big( f_\theta(x_{j,i}), y_{j,i} \big),$$

where $\ell \in [0, 1]$ is a bounded loss (e.g., 0–1 or a surrogate).

We define the *fairness gap* as the absolute difference in group risks:

$$\Delta L(f_\theta) := \big| L_0^{(s)}(f_\theta) - L_1^{(s)}(f_\theta) \big|, \quad \widehat{\Delta L}(f_\theta) := \big| \widehat{L}_0^{(s)}(f_\theta) - \widehat{L}_1^{(s)}(f_\theta) \big|.$$

This section develops three ingredients: (i) a discrepancy bound that relates the true and empirical group distributions, (ii) fairness guarantees for different fairness heads in the MTL objective, and (iii) excess-fairness bounds that connect optimisation error to fairness on unseen data.

## 4.1 DISCREPANCY GENERALISATION BOUND

We first control the discrepancy between group distributions, which serves as a proxy for fairness gaps. The following lemma shows that the true discrepancy is bounded by its empirical counterpart plus terms depending only on sample size and Rademacher complexity.

**Lemma 1** (Uniform control of discrepancy via Rademacher complexity). *Let $\mathcal{F} := \{x \mapsto \mathbf{1}\{h(x) \neq h'(x)\} : h, h' \in \mathcal{H}\}$. For dataset $D_j$, let*

$$\mathrm{Rad}_{D_j}(\mathcal{A}) = \frac{1}{n_j} \mathbb{E}_\sigma \left[ \sup_{a \in \mathcal{A}} \sum_{i=1}^{n_j} \sigma_i a(x_{j,i}) \right],$$

*where $\sigma_i$ are Rademacher variables. Then, with probability at least $1 - \delta$ over samples $D_0, D_1$,*

$$\mathrm{disc}(P_0, P_1) \leq \mathrm{disc}(\widehat{P}_0, \widehat{P}_1) + 2\,\mathrm{Rad}_{D_0}(\mathcal{F}) + 2\,\mathrm{Rad}_{D_1}(\mathcal{F}) + \sqrt{\tfrac{\ln(2/\delta)}{2n_0}} + \sqrt{\tfrac{\ln(2/\delta)}{2n_1}}. \quad (8)$$

This result implies that the observed discrepancy during training is a reliable proxy for the true hidden discrepancy, provided the hypothesis class is not overly complex and each group is sufficiently sampled. (See Appendix B for proof.)

**Lemma 2** (From loss-class to hypothesis-class complexity). *For binary-valued $\mathcal{H}$, $\mathrm{Rad}_{\mathcal{D}_j}(\mathcal{F}) \leq 2\,\mathrm{Rad}_{\mathcal{D}_j}(\mathcal{H})$.*

This lemma links the complexity of pairwise disagreements to the complexity of the original hypothesis class.

## 4.2 FAIRNESS HEADS IN THE MTL OBJECTIVE

We next show how the MTL objective enforces fairness through different fairness heads. Theorems 1 and 2 establish guarantees for adversarial and GLF losses, respectively.

**Theorem 1** (Fairness bound via adversarial head). *Let $\widehat{\varepsilon}(\theta)$ be the discriminator's best empirical error on the combined sample. Then, with probability at least $1 - \delta$, for every $\theta$,*

$$\Delta L(f_\theta) \leq 2 - 4\widehat{\varepsilon}(\theta) + 4\big(\mathrm{Rad}_{D_0}(\mathcal{H}) + \mathrm{Rad}_{D_1}(\mathcal{H})\big) + \sqrt{\tfrac{\ln(2/\delta)}{2n_0}} + \sqrt{\tfrac{\ln(2/\delta)}{2n_1}}.$$

In the MTL objective (Equation equation 3), instantiating $L_s$ as the adversarial discriminator loss minimises $\widehat{\varepsilon}(\theta)$, which in turn reduces the fairness gap $\Delta L(f_\theta)$. (Proof in Appendix D.)

**Theorem 2** (Fairness bound via GLF head). *For any $\theta$, with probability at least $1 - \delta$,*

$$\Delta L(f_\theta) \leq \widehat{\Delta L}(f_\theta) + 4\big(\mathrm{Rad}_{D_0}(\mathcal{H}) + \mathrm{Rad}_{D_1}(\mathcal{H})\big) + \sqrt{\tfrac{\ln(2/\delta)}{2n_0}} + \sqrt{\tfrac{\ln(2/\delta)}{2n_1}}.$$

This result formalises the effect of the GLF objective: by directly minimising the empirical fairness gap $\widehat{\Delta L}(f_\theta)$, the true fairness gap $\Delta L(f_\theta)$ is also reduced up to standard generalisation slack. (Proof in Appendix E.)

### 4.3 EXCESS-FAIRNESS BOUNDS

Finally, we connect optimisation error in the empirical MTL objective to fairness on unseen data. Let $\widehat{L}_{\text{MTL}}(\theta)$ denote the empirical counterpart of equation 3. Suppose $\hat{\theta}$ is an approximate empirical minimiser satisfying

$$\widehat{L}_{\text{MTL}}(\hat{\theta}) \leq \inf_{\theta} \widehat{L}_{\text{MTL}}(\theta) + \eta, \quad \eta \geq 0.$$

Let $\theta^{\star}$ be a reference (oracle) parameter set, and define the generalisation slack

$$C_{\text{gen}} := 4\big(\text{Rad}_{D_0}(\mathcal{H}) + \text{Rad}_{D_1}(\mathcal{H})\big) + \sqrt{\tfrac{\ln(2/\delta)}{2n_0}} + \sqrt{\tfrac{\ln(2/\delta)}{2n_1}}.$$

**Corollary 1** (Excess-fairness bound for GLF head). *If the fairness head implements the GLF penalty and $\hat{\theta}$ satisfies the above optimisation error, then with probability at least $1 - \delta$,*

$$\Delta L(f_{\hat{\theta}}) \leq \Delta L(f_{\theta^{\star}}) + \frac{\eta}{1 - \lambda} + 2C_{\text{gen}}.$$

**Corollary 2** (Excess-fairness bound for adversarial head). *If the fairness head is an adversarial discriminator, then with probability at least $1 - \delta$,*

$$\Delta L(f_{\hat{\theta}}) \leq \Delta L(f_{\theta^{\star}}) + \frac{4\eta}{1 - \lambda} + 2C_{\text{gen}}.$$

These corollaries demonstrate that small optimisation error translates into small fairness error, providing a convergence-to-fairness guarantee. (Proofs in Appendices F and G.)

## 5 EXPERIMENTAL SETUP

This section outlines our empirical evaluation setup of the proposed MTL approach. We begin by describing the dataset, then provide an overview of the baseline and comparative methods for benchmarking, and conclude with details on the experimental setup and evaluation protocol.

### 5.1 DATASETS

We evaluate fairness-aware classification using three standard benchmark datasets: Adult Income Becker & Kohavi (1996), Bank Marketing Moro & Cortez (2014), and COMPAS Angwin et al. (2016), as described in Le Quy et al. (2022). The Adult Income dataset is characterised by a binary classification task, namely the prediction of whether an individual earns more than $\$50,000$ per year. The sensitive attributes in consideration are 'gender' (binary) and 'race' (multi-class). The COMPAS dataset is centred on the classification of recidivism risk as a binary variable. The sensitive attributes in question are gender and race, as was the case in the previous dataset. The Bank Telemarketing Dataset focuses on the classification of samples with respect to subscription to a term bank deposit. The sensitive attribute in question is age, which is encoded into two groups: those who are considered to be 'privileged', with an age between 25 and 60 years old, and those who are considered to be 'unprivileged', with an age younger than 25 or older than 60 Le Quy et al. (2022).

We also evaluate fairness-aware classification using the Multi-Task Faces (MTF) image dataset Haffar (2024), which includes labelled face images for tasks such as face recognition, race, gender, and age classification. The curated version comprises 5,246 images that involve 240 unique individuals. The MTF datasets are ethically gathered, featuring publicly available images of celebrities and strictly adhering to copyright laws.

### 5.2 COMPARATIVE METHODS

We compare our proposed model architecture and loss functions with existing approaches to group-level fairness adaptation fairness-aware model design considering the Group-Specific Task Decomposition method Oneto et al. (2019), without applying any additional fairness constraints. As a regularisation-based method, we consider bounded group loss (BGL) Agarwal et al. (2019), Demographic parity loss (DPL), false positive rate loss (FPRL), and true positive rate loss (TPRL) proposed in Padala & Gujar (2021). For the adversarial debiasing method, we consider Fair adversarial discriminative Adel et al. (2019), an adversary debiasing min–max optimisation framework.

### 5.3 EXPERIMENTAL SETTINGS

Experiments are conducted using a 10-fold repeated hold-out protocol with an $0.6, 0.2, 0.2$ split for the training, validation and test sets, respectively. The model is trained for a maximum of 50 epochs with the best-performing epoch selected for the classification task on the validation set. Similarly to Zeng et al. (2024), the model architecture employs an $MTL$ framework with two fully connected layers in the shared representation and one task-specific fully connected layer per head. In line with previous studies, ReLU activations, dropout regularisation and stochastic gradient descent optimisation are employed. The same hyperparameter values adopted in Zeng et al. (2024) are used.

For the Computer Vision data experiments are conducted by fine-tuning a ConvNeXT Liu et al. (2022) model using the hyperparameters proposed in Haffar et al. (2025). The original ConvNeXT Liu et al. (2022) architecture is equipped with one task specific network for the classification and each sentitive task with a Layer Normalization Ba et al. (2016) and a fully connected layer. Our primary evaluation focuses on binary age classification (young vs old subjects).

The experiments evaluate fairness with respect to single sensitive attributes as well as multiple sensitive attributes concurrently.

### 5.4 EVALUATION METRICS

To evaluate classification performance, we use the *Area Under the Receiver Operating Characteristic Curve* (**AUC**). For fairness evaluation, we consider three well known group-based metrics the *Equal Opportunity Difference* (**EOpD**) Hardt et al. (2016), the *Equalised Odds Difference* (**EOD**) and the *Group AUC Difference* (**GAUCD**).

To jointly assess classification and fairness, we employ two aggregate metrics typical of MTL frameworks, the *Mean Rank* (**MR**) and *Delta-m %* ($\Delta_m\%$) Navon et al. (2022). The detailed formulas are reported in Appendix H.

## 6 RESULTS

In this section, we elaborate extensively on the empirical results obtained from our investigation. We perform a detailed comparison between the effectiveness of the $MTL$ framework proposed in our study and the various fairness-orientated loss functions that are being analysed. To provide further insight, we calculate the Pareto front, focussing on a single sensitive attribute, highlighting the trade-offs involved. When required, the hyperparameter $\lambda$ is set to $0.5$ to balance the trade-off

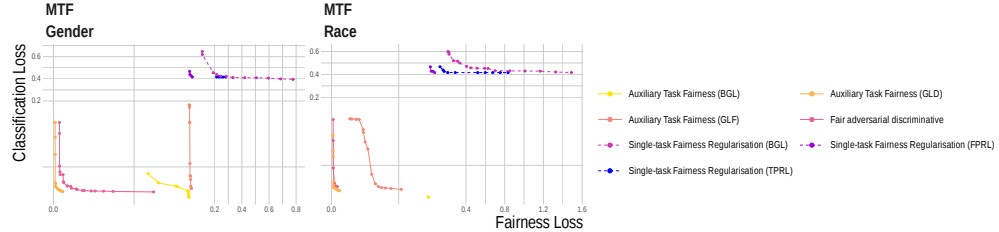

Figure 2: Pareto fronts on the MTF dataset for a single attribute.

between fairness and classification objectives, prioritising comparable performance across methodology rather than optimising them for the best overall accuracy. An ablation study to investigate the impact of the hyperparameter $\lambda$ in balancing the trade-off between fairness and accuracy is proposed in Appendix L. The evaluation includes two baseline models: a standard classification model that does not incorporate fairness constraints, referred to as *Single-task Classification*, and a fairness-aware model that applies established single-task regularisation techniques, denoted as *Single-task Fairness Regularisation*.

Figure 2 shows the Pareto fronts for the MTF dataset using one sensitive attribute, while Appendix K reports the Pareto fronts for tabular datasets. Our proposed method consistently identifies a superior front compared to baselines.

Tables 1 show our experimental results in data sets that involve two sensitive attributes. Our approach demonstrates effectiveness and scalability, showing no adverse effects from the number of sensitive attributes w.r.t. the *Single-task Classification* baseline model. Additional findings on tabular data sets related to individual sensitive attributes are detailed in Appendices J,I.

Table 1: Results on the different datasets with multiple sensitive attributes.

**Adult** — Race - Gender

| Architecture | Fairness Loss | Classification AUC↑ ± sd | Fairness Race EOD↓ ± sd | EopD↓ ± sd | GAUCD↓ ± sd | Fairness Gender EOD↓ ± sd | EopD↓ ± sd | GAUCD↓ ± sd | MR↓ | $\Delta_m\%$↓ |
|---|---|---|---|---|---|---|---|---|---|---|
| Single-task Classification (STL) | - | 89.082 ± 0.4 | 49.454 ± 3.56 | 80.143 ± 5.72 | 14.348 ± 5.63 | 6.335 ± 0.87 | 2.216 ± 1.86 | 5.846 ± 0.82 | 0 | 0 |
| Single-task Fairness Regularisation Agarwal et al. (2019) | BGL | 60.074 ± 16 | 54.529 ± 8.76 | 71.37 ± 7.75 | 24.753 ± 9.26 | 9.998 ± 7.21 | 7.968 ± 7.96 | 7.241 ± 6.8 | 6.875 | 239.11 |
| Single-task Fairness Regularisation Agarwal et al. (2019) | DPL | 57.129 ± 14.46 | 51.683 ± 11.74 | **69.343 ± 8.74** | 23.887 ± 10.66 | 9.824 ± 5.72 | 8.846 ± 8.18 | 7.527 ± 6.69 | 6.875 | 256.14 |
| Single-task Fairness Regularisation Agarwal et al. (2019) | FPRL | 57.129 ± 14.46 | 51.683 ± 11.74 | **69.343 ± 8.74** | 23.887 ± 10.66 | 9.824 ± 5.72 | 8.846 ± 8.18 | 7.527 ± 6.69 | 6.875 | 256.14 |
| Single-task Fairness Regularisation Agarwal et al. (2019) | TPRL | 57.129 ± 14.46 | 51.683 ± 11.74 | **69.343 ± 8.74** | 23.887 ± 10.66 | 9.824 ± 5.72 | 8.846 ± 8.18 | 7.527 ± 6.69 | 6.875 | 256.14 |
| Fair adversarial discriminative (min-max) Adel et al. (2019) | CE - BCE | 88.371 ± 0.45 | 49.605 ± 4.47 | 80.286 ± 5.98 | 15.832 ± 6.67 | 6.108 ± 1.64 | 2.009 ± 1.72 | 5.846 ± 1.04 | 2.75 | -0.25 |
| Group-Specific Task Decomposition Oneto et al. (2019) | BCE | 88.158 ± 0.45 | **48.882 ± 4.22** | 80.103 ± 5.74 | **14.036 ± 6.07** | 6.365 ± 1.24 | 2.149 ± 0.74 | **5.794 ± 1.01** | 3.5 | -2.37 |
| Auxiliary Task Fairness | BGL | **88.376 ± 0.46** | 50.346 ± 3.75 | 80.423 ± 5.62 | 15.93 ± 7.4 | 6.273 ± 1.74 | 1.966 ± 1.57 | 5.863 ± 1.03 | 3.75 | 1.39 |
| Auxiliary Task Fairness | GLD | 88.341 ± 0.44 | 50.144 ± 4.11 | 80.346 ± 5.5 | 16.031 ± 7.23 | 6.143 ± 1.72 | 1.748 ± 1.71 | 5.877 ± 1.04 | 4 | -4.28 |
| Auxiliary Task Fairness | GLF | 88.339 ± 0.44 | 50.096 ± 4.28 | 80.31 ± 5.66 | 15.906 ± 7.29 | **5.928 ± 1.5** | **1.631 ± 1.56** | 5.883 ± 1.05 | 3.5 | **-9.08** |

**Compas** — Race - Gender

| Architecture | Fairness Loss | Classification AUC↑ ± sd | Fairness Race EOD↓ ± sd | EopD↓ ± sd | GAUCD↓ ± sd | Fairness Gender EOD↓ ± sd | EopD↓ ± sd | GAUCD↓ ± sd | MR↓ | $\Delta_m\%$↓ |
|---|---|---|---|---|---|---|---|---|---|---|
| Single-task Classification (STL) | - | 95.082 ± 0.5 | 33.686 ± 15.19 | 55.63 ± 30.3 | 6.31 ± 1.68 | 4.621 ± 2.46 | 4.115 ± 4.1 | 1.106 ± 0.83 | 0 | 0 |
| Single-task Fairness Regularisation Agarwal et al. (2019) | BGL | 66.474 ± 20.9 | 56.567 ± 22.27 | 69.942 ± 24.91 | 19.168 ± 16.79 | 13.12 ± 9.84 | 11.434 ± 9.68 | 3.356 ± 2.87 | 7.875 | 461.35 |
| Single-task Fairness Regularisation Padala & Gujar (2021) | DPL | 67.844 ± 20.62 | 55.887 ± 19.61 | 70.39 ± 20.49 | 24.237 ± 20.85 | 12.064 ± 10.86 | 11.459 ± 10.53 | 3.032 ± 2.57 | 7.375 | 473.69 |
| Single-task Fairness Regularisation Padala & Gujar (2021) | FPRL | 67.844 ± 20.62 | 55.887 ± 19.61 | 70.39 ± 20.49 | 24.237 ± 20.85 | 12.064 ± 10.86 | 11.459 ± 10.53 | 3.032 ± 2.57 | 7.375 | 473.69 |
| Single-task Fairness Regularisation Padala & Gujar (2021) | TPRL | 67.844 ± 20.62 | 55.887 ± 19.61 | 70.39 ± 20.49 | 24.237 ± 20.85 | 12.064 ± 10.86 | 11.459 ± 10.53 | 3.032 ± 2.57 | 7.375 | 473.69 |
| Fair adversarial discriminative (min-max) Adel et al. (2019) | CE - BCE | 94.916 ± 0.52 | 34.213 ± 15.57 | 55.903 ± 31.2 | **6.337 ± 1.82** | 4.325 ± 1.93 | 3.134 ± 2.93 | **1.133 ± 0.88** | 2.875 | -12.53 |
| Group-Specific Task Decomposition Oneto et al. (2019) | BCE | 93.344 ± 0.91 | 34.559 ± 16.42 | **54.259 ± 31.23** | 9.164 ± 3.45 | **3.693 ± 2.09** | 3.358 ± 2.01 | 1.731 ± 1.13 | 3.75 | 33.51 |
| Auxiliary Task Fairness | BGL | 94.896 ± 0.56 | 33.459 ± 15.8 | 55.299 ± 31.62 | 6.696 ± 2.06 | 4.661 ± 2.26 | 4.405 ± 2.86 | 1.222 ± 0.85 | 3.75 | 11.77 |
| Auxiliary Task Fairness | GLD | **94.974 ± 0.53** | **33.391 ± 15.66** | 55.336 ± 31.54 | 6.601 ± 1.9 | 3.9 ± 1.37 | 3.365 ± 1.81 | 1.147 ± 0.93 | **2.25** | -13.36 |
| Auxiliary Task Fairness | GLF | 94.962 ± 0.52 | 34.11 ± 16.48 | 55.716 ± 32.02 | 6.517 ± 1.84 | 3.874 ± 1.43 | **3.104 ± 2.49** | 1.158 ± 0.91 | 2.375 | **-15.55** |

**Multi-Task Faces** — Race - Gender

| Architecture | Fairness Loss | Classification ACC↑ ± sd | Fairness Race EOD↓ ± sd | EopD↓ ± sd | GACCD↓ ± sd | Fairness Gender EOD↓ ± sd | EopD↓ ± sd | GACCD↓ ± sd | MR↓ | $\Delta_m\%$↓ |
|---|---|---|---|---|---|---|---|---|---|---|
| Single-task Classification (STL) | CE | 98.021 ± 1.34 | 17.923 ± 7.62 | 21.005 ± 9.08 | 13.226 ± 3.29 | 3.951 ± 2.26 | 4.793 ± 4.34 | 2.454 ± 3.28 | | |
| Single-task Fairness Regularisation Agarwal et al. (2019) | BGL | 80.484 ± 10.56 | 21.924 ± 7.5 | 24.605 ± 14.39 | 17.574 ± 4.86 | 4.258 ± 2.72 | 5.675 ± 3.31 | 10.019 ± 4.36 | 7.5 | 219.51 |
| Single-task Fairness Regularisation Padala & Gujar (2021) | DPL | 80.486 ± 9.85 | 22.15 ± 6.52 | 25.229 ± 14.11 | 17.043 ± 7.11 | 3.826 ± 2.65 | 4.698 ± 3.44 | 8.447 ± 5.94 | 7.375 | 173.78 |
| Single-task Fairness Regularisation Padala & Gujar (2021) | FPRL | 80.486 ± 9.85 | 22.15 ± 6.52 | 25.229 ± 14.11 | 17.043 ± 7.11 | 3.826 ± 2.65 | 4.698 ± 3.44 | 8.447 ± 5.94 | 7.375 | 173.78 |
| Single-task Fairness Regularisation Padala & Gujar (2021) | TPRL | 80.486 ± 9.85 | 22.15 ± 6.52 | 25.229 ± 14.11 | 17.043 ± 7.11 | 3.826 ± 2.65 | 4.698 ± 3.44 | 8.447 ± 5.94 | 7.375 | 173.78 |
| Fair adversarial discriminative (min-max) Adel et al. (2019) | BCE | 98.021 ± 0.39 | 16.236 ± 8.26 | 17.811 ± 11.13 | 12.63 ± 5.6 | 3.05 ± 2.64 | 3.987 ± 4.8 | 3.718 ± 3.33 | 3.25 | -8.80 |
| Group-Specific Task Decomposition Oneto et al. (2019) | BCE | 97.812 ± 0.38 | **15.663 ± 7.26** | **17.241 ± 8.46** | 10.245 ± 2.21 | 2.149 ± 0.74 | 2.559 ± 2.46 | 3.482 ± 3.79 | **1.375** | -50.74 |
| Auxiliary Task Fairness | BGL | 98.047 ± 0.45 | 21.294 ± 3.96 | 23.341 ± 6.6 | 12.239 ± 1.77 | 2.658 ± 2.05 | 3.352 ± 2.89 | 1.431 ± 0.74 | 2.875 | -9.26 |
| Auxiliary Task Fairness | GLD | **98.073 ± 0.44** | 17.137 ± 7.76 | 19.86 ± 8.9 | 11.938 ± 3.35 | 3.211 ± 1.58 | 3.238 ± 2.19 | **0.667 ± 0.78** | 3.375 | **-71.85** |
| Auxiliary Task Fairness | GLF | 97.917 ± 0.32 | 18.701 ± 4.6 | 23.1 ± 5.98 | 12.772 ± 0.49 | 3.133 ± 2.14 | 4.976 ± 4 | 3.449 ± 3.86 | 4.5 | 17.1 |

The results consistently show that treating fairness as an auxiliary task yields more balanced outcomes compared to single-task regularisation and adversarial debiasing. Across datasets, our proposed GLF and GLD losses reduce group disparities while preserving accuracy, validating our theoretical guarantees. Importantly, the framework generalises effectively to multiple sensitive attributes and diverse modalities. Together, these findings suggest that multi-task fairness is a robust and generalisable paradigm for fair representation learning.

## 7 CONCLUSIONS

In group-level fairness training, single-task models with fairness regularisation typically encounter an undesirable trade-off between preserving predictive accuracy and ensuring fairness. This study explores the potential of $MTL$ to mitigate this trade-off by introducing Auxiliary Task Fairness, a novel $MTL$ architecture that treats fairness as an auxiliary task with dedicated task-specific layers and fairness losses. Our proposed framework enables the concurrent learning of predictive and fairness objectives by guiding the shared representation to be both accurate and unbiased with respect to sensitive attributes, while maintaining task-specific layers to enhance prediction performance and fairness awareness. Theoretical analysis demonstrates convergence-to-fairness guarantees. Extensive empirical evaluations across benchmark datasets show that Auxiliary Task Fairness effectively improves the balance between predictive performance and group-level fairness, outperforming conventional regularisation-based methods and adversarial debiasing methods. Future work includes expanding the comparative analysis to incorporate a broader range of fairness loss functions and extending the architecture to handle multiple fairness definitions simultaneously.

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
