## A  SETUP NOTATION

| Symbol | Meaning |
|--------|---------|
| $\mathcal{L}_{\mathrm{MTL}}$ | Multi-task learning objective |
| $\mathcal{H}$ | Hypothesis class |
| disc | Distribution discrepancy |
| $\mathrm{Rad}_{\mathcal{D}}(\mathcal{H})$ | Rademacher complexity on dataset $\mathcal{D}$ |
| $\mathcal{L}_j^{(s)}(f)$ | True group-$j$ loss of $f$ |
| $\widehat{\mathcal{L}}_j^{(s)}(f)$ | Empirical group-$j$ loss of $f$ |
| $\Delta L(f)$ | True fairness gap |
| $\widehat{\Delta L}(f)$ | Empirical fairness gap |
| $\widehat{\varepsilon}(\theta)$ | Empirical adversary error |
| $C_{\mathrm{gen}}$ | generalisation slack term |
| $\theta^{\star}$ | Oracle parameter |
| $z_\theta(x)$ | Latent embedding of input $x$ |
| $\hat{P}_0^\theta, \hat{P}_1^\theta$ | Empirical latent distributions for groups 0 and 1 |
| $\hat{\mu}_0, \hat{\mu}_1$ | Empirical latent means |
| $\hat{\Sigma}_0, \hat{\Sigma}_1$ | Empirical latent covariances |
| $\epsilon$ | Regularisation for invertibility |
| $L_s^{\mathrm{KL}}(\theta)$ | KL-based fairness loss |
| $\varepsilon_{\mathrm{KL}}(n, d, \delta)$ | Empirical KL concentration error |
| $\hat{\theta}_t$ | SGD iterate at step $t$ |
| $H_{\max}$ | Uniform Hessian spectral norm bound |
| $\mathcal{F}_h$ | Fairness function class on latent embeddings |

Table 2: Notation for multi-task fairness generalisation analysis, extended with KL-based fairness symbols.

## B  PROOF: UNIFORM CONTROL OF DISCREPANCY VIA RADEMACHER COMPLEXITY

*Proof.* By definition of discrepancy,

$$\mathrm{disc}(P_0, P_1) \overset{(a)}{\leq} \mathrm{disc}(P_0, \widehat{P}_0) + \mathrm{disc}(\widehat{P}_0, \widehat{P}_1) + \mathrm{disc}(\widehat{P}_1, P_1)$$

(a) follows directly from the triangle inequality for the distribution discrepancy.

For each $j \in \{0, 1\}$, consider the uniform deviation

$$\sup_{f \in \mathcal{F}} \left| \mathbb{E}_{P_j}[f] - \widehat{\mathbb{E}}_{\mathcal{D}_j}[f] \right| \overset{(b)}{\leq} 2\,\mathrm{Rad}_{\mathcal{D}_j}(\mathcal{F}) + \sqrt{\frac{\ln(2/\delta)}{2n_j}},$$

where (b) follows from symmetrisation and McDiarmid's inequality.

Applying the union bound over $j = 0, 1$ and inserting into the previous inequality gives the stated lemma. $\square$

## C  PROOF: FROM LOSS-CLASS TO HYPOTHESIS-CLASS COMPLEXITY

*Proof.* Using the identity $\mathbf{1}\{h \neq h'\} = \frac{1}{2}(1 - hh')$ and the Ledoux–Talagrand contraction (or direct symmetrisation on products), we obtain the factor 2 relating the Rademacher complexities of the loss class and the hypothesis class. $\square$

## D    PROOF: MULTI-TASK FAIRNESS BOUND

*Proof.* For any $f_\theta$, we have

$$\Delta L(f_\theta) \overset{(a)}{\leq} \mathrm{disc}(P_0, P_1),$$

where (a) follows from the fact that the discrepancy is a supremum over pairs in $\mathcal{H}$.

By Lemma 1 and Lemma 2:

$$\mathrm{disc}(P_0, P_1) \overset{(b)}{\leq} \mathrm{disc}(\widehat{P}_0, \widehat{P}_1) + 4\big(\mathrm{Rad}_{\mathcal{D}_0}(\mathcal{H}) + \mathrm{Rad}_{\mathcal{D}_1}(\mathcal{H})\big)$$
$$+ \sqrt{\frac{\ln(2/\delta)}{2n_0}} + \sqrt{\frac{\ln(2/\delta)}{2n_1}},$$

(b) follows from the uniform deviation bounds and union bound over $j = 0, 1$.

Finally, for empirical distributions:

$$\mathrm{disc}(\widehat{P}_0, \widehat{P}_1) \overset{(c)}{=} 2\big(1 - 2\widehat{\varepsilon}(\theta)\big),$$

(c) follows from the definition of the empirical $\mathcal{H}$-divergence Ben-David et al. (2010); Mansour et al. (2009).

Combining all displays yields the theorem statement. □

## E    PROOF: GLF MULTI-TASK FAIRNESS BOUND

*Proof.* For any $h \in \mathcal{H}$, define $f_h(x, y) := \ell(h(x), y)$ and let $\mathcal{G} := \{f_h : h \in \mathcal{H}\} \subset [0, 1]^{\mathcal{X} \times \mathcal{Y}}$.

By symmetrisation and McDiarmid's inequality:

$$\big|\mathcal{L}_j^{(s)}(h) - \widehat{\mathcal{L}}_j^{(s)}(h)\big| \overset{(a)}{\leq} 2\,\mathrm{Rad}_{\mathcal{D}_j}(\mathcal{G}) + \sqrt{\frac{\ln(2/\delta)}{2n_j}},$$

(a) follows from uniform convergence arguments for bounded losses.

By the contraction lemma Bartlett & Mendelson (2002), $\mathrm{Rad}_{\mathcal{D}_j}(\mathcal{G}) \leq \mathrm{Rad}_{\mathcal{D}_j}(\mathcal{H})$. Applying the triangle inequality:

$$\Delta L(h) \overset{(b)}{\leq} \widehat{\Delta} L(h) + \sum_{j=0}^{1} \big|\mathcal{L}_j^{(s)}(h) - \widehat{\mathcal{L}}_j^{(s)}(h)\big|,$$

(b) follows directly from $|a - b| \leq |a - c| + |c - b|$.

Substituting the uniform bounds for $j = 0, 1$ gives the theorem. □

## F    PROOF: EXCESS-FAIRNESS BOUND FOR GLF HEAD (F2)

*Proof.* Decompose the empirical MTL objective:

$$\widehat{\mathcal{L}}_{\mathrm{MTL}}(\theta) = \lambda\,\widehat{\mathcal{T}}(\theta) + (1 - \lambda)\,\widehat{\mathcal{F}}(\theta),$$

where $\widehat{\mathcal{F}}(\theta) = \widehat{\Delta} L(f_\theta)$.

Empirical suboptimality implies:

$$\widehat{\Delta} L(f_{\widehat{\theta}}) \overset{(a)}{\leq} \widehat{\Delta} L(f_{\theta^\star}) + \frac{\eta}{1 - \lambda},$$

(a) follows from rearranging $(1 - \lambda)\widehat{\Delta} L(f_{\widehat{\theta}}) \leq (1 - \lambda)\widehat{\Delta} L(f_{\theta^\star}) + \eta$.

Applying Theorem 2 to both $\widehat{\theta}$ and $\theta^\star$:

$$\Delta L(f_{\widehat{\theta}}) \overset{(b)}{\leq} \widehat{\Delta}L(f_{\widehat{\theta}}) + C_{\text{gen}},$$

$$\Delta L(f_{\theta^\star}) \overset{(c)}{\geq} \widehat{\Delta}L(f_{\theta^\star}) - C_{\text{gen}},$$

(b) and (c) follow from the generalisation bound. Combining inequalities:

$$\Delta L(f_{\widehat{\theta}}) \overset{(d)}{\leq} \Delta L(f_{\theta^\star}) + \frac{\eta}{1-\lambda} + 2C_{\text{gen}},$$

(d) follows from chaining the previous steps. $\square$

## G    PROOF: EXCESS-FAIRNESS BOUND FOR ADVERSARIAL HEAD (F1)

*Proof.* Empirical suboptimality implies:

$$\widehat{\varepsilon}(\widehat{\theta}) \overset{(a)}{\leq} \widehat{\varepsilon}(\theta^\star) + \frac{\eta}{1-\lambda},$$

(a) follows directly from the F1 objective.

By Theorem 1:

$$\Delta L(f_\theta) \leq 2 - 4\widehat{\varepsilon}(\theta) + C_{\text{gen}}.$$

Apply to $\widehat{\theta}$ and $\theta^\star$:

$$\begin{aligned}
\Delta L(f_{\widehat{\theta}}) &\overset{(b)}{\leq} 2 - 4\widehat{\varepsilon}(\widehat{\theta}) + C_{\text{gen}} \\
&\overset{(c)}{\leq} 2 - 4\widehat{\varepsilon}(\theta^\star) - \frac{4\eta}{1-\lambda} + C_{\text{gen}} \\
&\overset{(d)}{=} \Delta L(f_{\theta^\star}) + \frac{4\eta}{1-\lambda} + 2C_{\text{gen}},
\end{aligned}$$

where (b) follows from Theorem 1, (c) from the empirical suboptimality bound, and (d) from rearranging $\Delta L(f_{\theta^\star}) \leq 2 - 4\widehat{\varepsilon}(\theta^\star) + C_{\text{gen}}$. $\square$

## H    EVALUATION METRICS

To evaluate classification performance, we use the *Area Under the Receiver Operating Characteristic Curve* (**AUC**). For fairness evaluation, we consider three well known group-based metrics. The *Equal Opportunity Difference* (**EOpD**) Hardt et al. (2016) measures the maximum disparity in true positive rates (TPR) across sensitive groups . Given $G$ groups, let $\text{TPR}_g$ be the true positive rate for group $g$. Then the metric is define in equation 9.

$$\text{EOpD} = \max_{g \in G} \text{TPR}_g - \min_{g \in G} \text{TPR}_g \tag{9}$$

The *Equalised Odds Difference* (**EOD**) extends EOpD by also considering false positive rates (FPR). It is defined in equation 10 the average of the maximum inter-group disparities in TPR and FPR.

$$\text{EOD} = \frac{1}{2}\left( \max_{g \in G} \text{TPR}_g - \min_{g \in G} \text{TPR}_g + \max_{g \in G} \text{FPR}_g - \min_{g \in G} \text{FPR}_g \right) \tag{10}$$

The *Group AUC Difference* (**GAUCD**) captures disparities in classification performance across sensitive groups. Given group-wise AUC scores, it is defined in equation 11.

$$\text{GAUCD} = \max_{g \in G} \text{AUC}_g - \min_{g \in G} \text{AUC}_g \tag{11}$$

where $\text{AUC}_g$ is the AUC for group $g$, and $G$ is the set of all sensitive groups.

To jointly assess classification and fairness, we employ two aggregate metrics. The *Mean Rank* **(MR)** computes the average rank of each method across all evaluation metrics, as defined in equation 12.

$$\text{MR} = \frac{1}{M} \sum_{i=1}^{M} \text{rank}_i \tag{12}$$

where $M$ is the number of metrics and $\text{rank}_i$ is the rank assigned for metric $i$.

The *Delta-m %* ($\Delta_m\%$) score quantifies the relative improvement of a $MTL$ model over the $STL$, as detailed in equation 13.

$$\Delta m\% = \frac{1}{K} \left[ \sum_{k \in \text{accuracy}} (-1)^{\nu_k} \frac{1}{N_{acc}} \frac{m_{mtl,k} - m_{stl,k}}{m_{stl,k}} + \sum_{k \in \text{fairness}} (-1)^{\nu_k} \frac{1}{N_f} \frac{m_{mtl,k} - m_{stl,k}}{m_{stl,k}} \right] \cdot 100 \tag{13}$$

Here, $m_{mtl,k}$ and $m_{stl,k}$ denote the performance of the $k$-th metric for the $MTL$ and $STL$ models, respectively. The formula accounts for metrics scaled by the reciprocal of the number of precision metrics, $1/N_{acc}$, fairness metrics, $1/N_f$, to ensure that their contributions are normalised. The binary indicator $\nu_k$ adjusts the sign of each term, with $\nu_k = 1$ when higher values indicate better performance (e.g. accuracy) and $\nu_k = 0$ when lower values are preferable (e.g., error). This combined weighting ensures that $\Delta_m\%$ provides a balanced measure of both predictive performance and fairness improvements in a single score.

# I RESULTS ON COMPUTER VISION DATASET WITH ONE SENSITIVE ATTRIBUTE

Table 3: Results on the Computer Vision Dataset with specific sensitive attribute.

**Multi-Task Faces** — *Gender*

| Architecture | Fairness Loss | Classification AUC↑ ± sd | Fairness EOD↓ ± sd | EopD↓ ± sd | GAUCD↓ ± sd | MR↓ | $\Delta_m\%$↓ |
|---|---|---|---|---|---|---|---|
| Single-task Classification (STL) | - | 97.688 ± 0.14 | 3.385 ± 2.55 | 4.496 ± 4.09 | 4.71 ± 3.56 | | |
| Single-task Fairness Regularisation Agarwal et al. (2019) | BGL | 81.759 ± 10.05 | 5.531 ± 0.92 | 7.205 ± 2.67 | 10.705 ± 4.47 | 9 | 99.96 |
| Single-task Fairness Regularisation Padala & Gujar (2021) | DPL | 85.803 ± 5.34 | 4.65 ± 1.65 | 5.481 ± 2.69 | 10.179 ± 5.47 | 7 | 70.63 |
| Single-task Fairness Regularisation Padala & Gujar (2021) | FPRL | 85.803 ± 5.34 | 4.65 ± 1.65 | 5.481 ± 2.69 | 10.179 ± 5.47 | 7 | 70.63 |
| Single-task Fairness Regularisation Padala & Gujar (2021) | TPRL | 85.803 ± 5.34 | 4.65 ± 1.65 | 5.481 ± 2.69 | 10.179 ± 5.47 | 7 | 70.63 |
| Fair adversarial discriminative (min-max) Adel et al. (2019) | BCE | **97.766** ± 0.25 | 3.663 ± 0.8 | 4.953 ± 3.2 | 3.247 ± 3.69 | 3.75 | -4.3 |
| Group-Specific Task Decomposition Oneto et al. (2019) | BCE | 97.609 ± 0.24 | 3.431 ± 1.39 | 4.662 ± 1.34 | 1.428 ± 0.68 | 3.375 | -21.46 |
| Auxiliary Task Fairness | BGL | 97.734 ± 0.15 | 2.832 ± 1.73 | 3.195 ± 2.94 | 3.297 ± 3.34 | 2.75 | -25.14 |
| Auxiliary Task Fairness | GLD | 97.672 ± 0.17 | 3.165 ± 2.86 | 4.131 ± 4.7 | 3.239 ± 4.07 | 3 | -15.27 |
| Auxiliary Task Fairness | GLF | 97.609 ± 0.18 | **1.926** ± 1.64 | **2.408** ± 1.4 | **1.676** ± 2.32 | **2.125** | **-51.24** |

**Multi-Task Faces** — *Race*

| Architecture | Fairness Loss | Classification AUC↑ ± sd | Fairness EOD↓ ± sd | EopD↓ ± sd | GAUCD↓ ± sd | MR↓ | $\Delta_m\%$↓ |
|---|---|---|---|---|---|---|---|
| Single-task Classification (STL) | - | 97.688 ± 0.14 | 15.695 ± 6.99 | 17.68 ± 8.21 | 11.44 ± 2.57 | | |
| Single-task Fairness Regularisation Agarwal et al. (2019) | BGL | 83.63 ± 6.02 | 22.967 ± 7.05 | 26.253 ± 11.43 | 18.006 ± 3.98 | 7.5 | 65.13 |
| Single-task Fairness Regularisation Padala & Gujar (2021) | DPL | 89.054 ± NA | 25.793 ± NA | 31.795 ± NA | 16.648 ± NA | 7.5 | 72.07 |
| Single-task Fairness Regularisation Padala & Gujar (2021) | FPRL | 89.054 ± NA | 25.793 ± NA | 31.795 ± NA | 16.648 ± NA | 7.5 | 72.07 |
| Single-task Fairness RegularisationPadala & Gujar (2021) | TPRL | 85.803 ± 5.34 | 22.634 ± 6.63 | 26.929 ± 13.6 | 18.578 ± 5.11 | 7.5 | 65.14 |
| Fair adversarial discriminative (min-max) Adel et al. (2019) | BCE | 97.766 ± 0.3 | 16.628 ± 8.12 | 18.896 ± 8.99 | 11.199 ± 2.98 | 1.75 | 3.5 |
| Group-Specific Task Decomposition Oneto et al. (2019) | CE | 97.609 ± 0.42 | 20.167 ± 7.65 | 23.131 ± 10.84 | 13.48 ± 5.61 | 4.5 | 25.8 |
| Auxiliary Task Fairness | BGL | 97.578 ± 0.22 | 18.14 ± 7.09 | 20.544 ± 9.49 | **10.979** ± 3.08 | 2.75 | 9.36 |
| Auxiliary Task Fairness | GLD | **97.703** ± 0.19 | **15.695** ± 7.82 | **17.68** ± 9.17 | 11.44 ± 2.87 | **1.75** | **-0.02** |
| Auxiliary Task Fairness | GLF | 97.563 ± 0.45 | 19.235 ± 6.42 | 22.623 ± 9.53 | 12.9 ± 4.59 | 4.25 | 21.22 |

## J RESULTS ON TABULAR DATASETS WITH ONE SENSITIVE ATTRIBUTE

Table 4: Results on the different dataset with specific sensitive attribute.

**Adult**      *Race*

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

**COMPAS**      *Gender*

| Architecture | Fairness Loss | Classification AUC↑ ± sd | Fairness EOD↓ ± sd | EopD↓ ± sd | GAUCD↓ ± sd | MR↓ | $\Delta_m$%↓ |
|---|---|---|---|---|---|---|---|
| Single-task Classification (STL) | - | 70.58 ± 25.23 | 10.51 ± 13.08 | 8.37 ± 10.13 | 3.31 ± 3.58 | | |
| Single-task Fairness Regularisation Agarwal et al. (2019) | BGL | 69.718 ± 17.8 | 12.066 ± 13.38 | 11.093 ± 11.35 | 3.416 ± 3.55 | 6.75 | 18.033 |
| Single-task Fairness Regularisation Padala & Gujar (2021) | DPL | 69.507 ± 17.71 | 12.654 ± 13.2 | 11.088 ± 11.39 | 3.454 ± 3.58 | 7.75 | 20.55 |
| Single-task Fairness Regularisation Padala & Gujar (2021) | FPRL | 69.507 ± 17.71 | 12.654 ± 13.2 | 11.088 ± 11.39 | 3.454 ± 3.58 | 7.75 | 20.55 |
| Single-task Fairness Regularisation Padala & Gujar (2021) | TPRL | 69.507 ± 17.71 | 12.654 ± 13.2 | 11.088 ± 11.39 | 3.454 ± 3.58 | 7.75 | 20.55 |
| Group-Specific Task Decomposition Oneto et al. (2019) | BCE | 93.413 ± 7.41 | 4.509 ± 3.27 | 3.779 ± 3.36 | 2.13 ± 1.81 | 4.5 | -81.56 |
| Fair adversarial discriminative (min-max) Adel et al. (2019) | CE | **94.389** ± 7.81 | **3.356** ± 1.93 | 3.78 ± 3.29 | 1.558 ± 0.94 | 2.5 | -92.36 |
| Auxiliary Task Fairness | BGL | 93.678 ± 11.19 | 3.404 ± 2.21 | 3.301 ± 3.99 | 1.397 ± 1.12 | **2** | **-94.73** |
| Auxiliary Task Fairness | GLF | 93.551 ± 11.12 | 3.444 ± 2.37 | 4.277 ± 3.73 | **1.202** ± 1.05 | 3.25 | -92.50 |
| Auxiliary Task Fairness | GLD | 93.676 ± 11.19 | 3.531 ± 2.49 | **3.219** ± 4.09 | 1.405 ± 1.12 | 2.75 | -94.57 |

## K    PARETO FRONT FOR SINGLE SENSITIVE ATTRIBUTE

Figure 3 shows the Pareto fronts for the tabular datasets with a single attribute. Our proposed method consistently identifies a superior front compared to baselines.

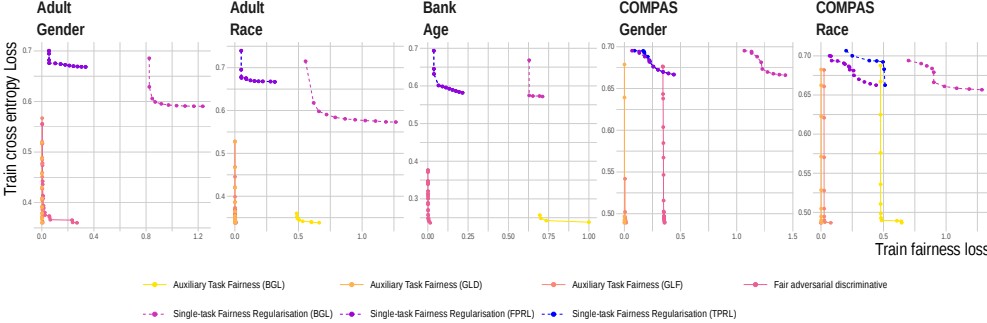

Figure 3: Pareto fronts on the tabular datasets for a single attribute

## L    SENSITIVITY STUDY FOR THE HYPERPARAMETER $\lambda$

The hyperparameter $\lambda$ modulates the trade-off between accuracy and fairness. However, its utility is model-specific, with effects contingent on the underlying architecture. In *Group-Specific Task Decomposition*, adjusting $\lambda$ alters group prioritisation rather than the direct accuracy-fairness trade-off. Conversely, in *Auxiliary Task Fairness* and *Fair adversarial discriminative* models, $\lambda$ assigns weights to task-specific layers, directly impacting gradient updates, as depicted in Figure 1. In the *Single-task Fairness Regularisation* method, $\lambda$ explicitly mediates between classification loss and the fairness regularisation term.

Figure 4 presents the classification and fairness performance indices against hyperparameter values for the three tabular datasets. The data series denote the tested models across different architectures. In the first column plots (AUC), *Single-task Fairness Regularisation* models (dashed lines) surpass $MTL$ architectures in classification regardless of $\lambda$. The second column displays (-EOD) results, where higher values reflect enhanced fairness. Here, increasing the hyperparameter does not improve fairness in *Single-task Fairness Regularisation* models, unlike in $MTL$ approaches which do improve. The third column contrasts classification performance with group fairness, highlighting models with better trade-offs. The *Auxiliary Task Fairness* model adeptly manages the balance between these objectives across varying $\lambda$. It often delivers top-tier performance among all models, corroborating findings in Table 4.

## M    QUALITATIVE ANALYSIS OF THE LATENT REPRESENTATION $h$

The core objective of the proposed *Auxiliary Task Fairness* architecture is to learn a shared latent representation $h$ that is unbiased to sensitive attributes while remaining predictive for the main task. To evaluate the degree of this bias qualitatively, we perform a t-SNE van der Maaten & Hinton (2008) dimensionality reduction of the latent representation $h$ of the last shared layer. The visual inspection of the top two dimensions allows to analyse the latent space and assess whether the *Auxiliary Task Fairness* model reduces sensitivity to the protected attribute.

Figure 5 illustrates the 2D t-SNE embeddings for the *Single-task Fairness Regularisation* (BGL loss) and the *Auxiliary Task Fairness* (GLF loss) models. Each column represents results for 5 values of hyperparameter $\lambda$. Density plots show sensitive group distributions, and scatter points indicate test outcomes. At $\lambda = 0.1$, the emphasis is on fairness rather than classification, leading to high error concentration. At $\lambda = 0.9$, prioritising classification improves accuracy with fewer

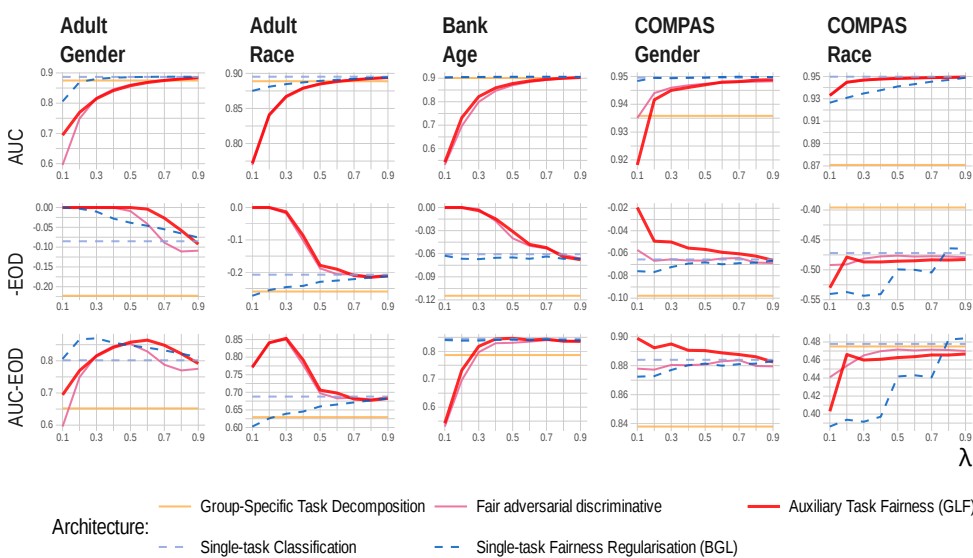

Figure 4: Trend of classification performance ($AUC$) and fairness performance ($EOD$) as the parameter $\lambda$ varies, evaluated across different datasets and sensitive attributes. The interaction between the two metrics ($AUC \cdot EOD$) reflects the ability of the model to maintain classification accuracy while reducing bias.

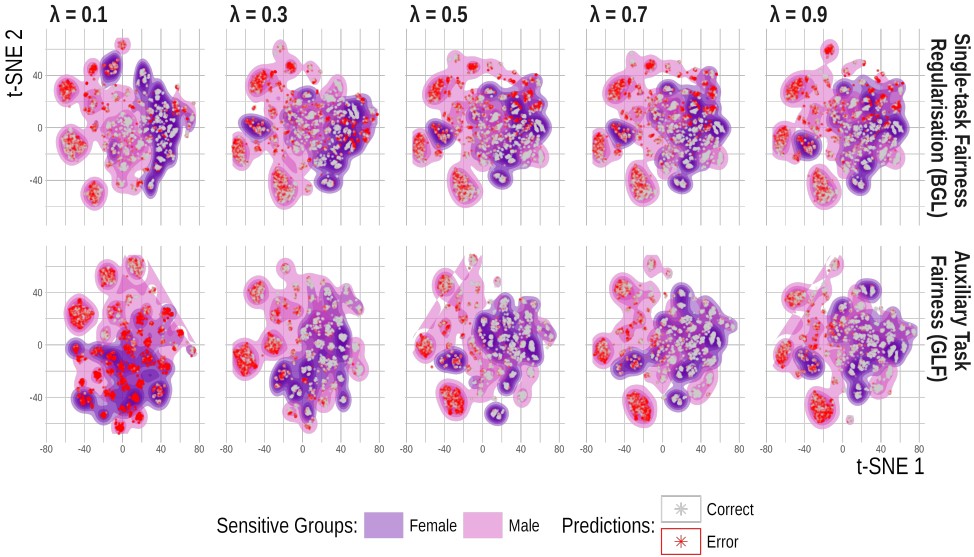

Figure 5: t-SNE plots of the latent representations from the last shared layer of the models trained on the Adult dataset, using *Gender* as the sensitive attribute. The effect of varying the fairness weight $\lambda$ is shown. Density plots represent the distribution of sensitive groups over the first two t-SNE components, while individual points are coloured by classification outcomes (confusion matrix) using a threshold to maximise AUC score on the validation data set.

errors. Figures indicate poor performance for $\lambda = 0.1$ due to insufficient supervision. Increasing $\lambda$ enhances classification focus and accuracy, as shown in Fig. 4 with increasing AUC. Notably, the GLF model distributes errors across space with reduced bias, while the BGL model exhibits error overlap with unprivileged groups, indicating higher bias. Thus, the *Auxiliary Task Fairness* approach appears to better disentangle sensitive attributes, maintaining predictive success.