# OpenReview forum: "Learning fair latent representation with Multi-Task Deep Learning"
_ICLR.cc/2026/Conference — Submitted to ICLR 2026_

### Official Review · Reviewer_6fnZ · 2025-10-23

**Soundness:** 3
**Presentation:** 3
**Contribution:** 2
**Rating:** 4
**Confidence:** 4

**Summary:**

This work aims to learn fair latent representation through multi-objective optimisation in a multi-task learning (MTL) paradigm. As such, two novel fairness loss functions are proposed for model training under the MTL approach. A theoretical analysis of the generalisation properties and empirical results supports the proposed method.

Overall, the MTL paradigm shows existing work. However, it is unclear if the proposed loss term is effective (compared to BGL). Most improvement comes from the MTL paradigm. The authors are encouraged to clarify this concern.

**Strengths:**

- This work considers fair representation as a multi-task learning problem, where group fairness is regarded as an auxiliary task to augment the main prediction task. Under this framework, two new fairness-oriented loss functions are proposed to guide model optimisation. The latest loss function aims to minimise the loss between different sensitive groups and promote equitable treatment across sensitive groups.
- This work provided a theoretical analysis to show how multi-task objective provably reduces the actual group fairness gap via explicit generalisation bounds. It also establishes rigorous optimisation-to-fairness guarantees in a multi-task learning paradigm. I did not find obvious errors in the theoretical proof.
- Code is provided (anonymously) for review and will (probably) be made available in the future.

**Weaknesses:**

According to the empirical results, the adaptation of MTL for fair latent representation learning is effective. However, it is not entirely clear how the two loss terms, GLD and GLF, contribute significantly to MTL. The difference in performance is significantly smaller than the SD. The author is encouraged to conduct a statistical analysis to justify the proposed loss term.

**Questions:**

- Please discuss why in COMPAS, the performance of all three compared fairness losses are identical.
- In the experiment, models from the best-performing epoch are selected for results reporting. Can the author confirm if the training stabilises at certain epochs, or experiences high fluctuation? Also, do you know if the selected epoch is consistent across the 10-fold validation? It would be problematic if the most optimal performance (in terms of epoch) is chosen on each hold-out set.
- Missing related work: (1) Fair Representation: Guaranteeing Approximate Multiple Group Fairness for Unknown Tasks, TPAMI 2023, (2) FARE: Provably Fair Representation Learning with Practical Certificates, ICML 2023.

Minor comments:
- In Fig. 5, the colour label between the female and male groups is very close; it is hard to interpret the region in the plot.
- Table 1 is overly small for print.
- In Fig. 3 (supplementary material), the label is unreadable. Authors should always consider readability in print.
- The citation is not appropriately referenced in the main text.
- 10-fold repeated hold-out protocol -> 10-fold cross-validation

---

> ### Author Response · Authors · 2025-11-21
>
> We acknowledge that the empirical analysis could be further strengthened to isolate the source of performance improvements more clearly.
>
> We confirm that learning stabilises during training with no significant fluctuations.
> Additionally, epochs of best models remain consistent throughout the hold-out experimental process, which is a 10-time repeated hold-out method.
>
> The significant contribution of this paper is indeed the explicit formulation of fair representation learning as a multi-task learning problem.
> Our newly introduced fairness loss functions are uniquely tailored for this framework. Unlike traditional performance-based metrics directly applied as regularisation term to the data loss function, our method requires a loss function as an independent auxiliary task.
> We agree that it is unnecessary to propose two options, if there is no significant difference between them.
>
> We thank the reviewer for pointing out these related works; these are valuable references and we will include and discuss them.

---

> > ### Comment · Reviewer_6fnZ · 2025-11-26
> > **Acknowledgement of rebuttal**
> >
> > I sincerely thank you for the authors' response. The response addressed the concerns I had in the initial review.
> >
> > I will carefully consider the response, the submitted manuscript, and the fellow reviewers' assessments in making the final decision.

---

### Official Review · Reviewer_7nuJ · 2025-10-30

**Soundness:** 1
**Presentation:** 3
**Contribution:** 1
**Rating:** 2
**Confidence:** 4

**Summary:**

This paper presents fairness training with Multi-task learning (MTL). Compared to single-task learning, the author claims that using fairness loss in MTL can achieve a better trade-off performance. The author proposed two fairness losses as different tasks to train the backbone model, such that the group performance can be more even. The methods are being tested on the benchmark fairness-related research widely accepted datasets, and the author claims that their method outperforms the baselines.

**Strengths:**

1. MTL has recently been a hot research topic.

2. This paper has made a great effort to find a theoretical analysis of the generalization properties of MTL.

3. Experiments result are trustworthy.

**Weaknesses:**

I have read this paper at least three times.

Unfortunately, this paper has a fundamental flaw in modeling fairness. To put it briefly, this paper underestimated the approach to achieving fairness. **Making the losses from different demographic groups equal or similar is not an appropriate way to ensure fairness.**

No matter from Equation (5), Equation (6), or line 274, we can see that the authors are trying to make different sensitive groups' losses similar, e.g., the performance of males and females to be the same. However, one of the major causes that forms fairness issues is spurious correlation. A spurious correlation occurs when the task label correlates with the sensitive attribute but has no causal effect. A very famous example is Waterbirds. The task is to identify the type of bird (waterbird or landbird), and the sensitive attribute is the background (water background and land background). The spurious relationship is that the waterbird with a water background has a much higher chance than the waterbird with a land background, and vice versa. Because there are data imbalance issues within a sensitive attribute, if we use the loss function presented in equation (5), which asks for the performance of the water background and the land background show the similiar loss, the advantage groups (waterbird with water background and landbird with land background) will dominate and achieve similiar good perfornace, and two disadvantage groups (waterbird with land background and landbird with water background) show similar bad performance. But it also meets the constraints that two sensitive groups have similar losses. If you compute Equal Opportunity Difference, there would be a large value.

That explains why the results from Table 1 show very limited or no improvement compared with the very weak baseline Single-task Classification (STL)

I would highly recommend that the author read this section and read papers like GroupDRO to refine their methods. A straightforward way to modify your methods is to define the group as the Cartesian product between the sensitive attribute and the target attribute.

Beyond the fundamental flaw, I would also like to point out an issue in the theorem section. Since your claim is that fairness in MTL is better than in STL with a regularizer, I would like to see a theoretical aspect that proves this point.

In addition, this method shows no advantage over existing approaches in terms of annotation requirements. For example, it still requires both target labels and sensitive labels, and it also needs to be trained from scratch. Therefore, the motivation for accepting this paper is reduced, especially considering that there is already an abundance of research on fairness.

**Questions:**

I would like to apologize for not being able to vote to accept this paper, but I hope my comments can help improve your submission in the future. I don't have more questions.

---

> ### Author Response · Authors · 2025-11-21
>
> We thank the reviewer for the insightful example and and for attention dedicated to our paper. We fully agree that the issue of spurious correlations is important. However, we believe that the shortcoming identified in the Waterbird example arises from a mismatch between the problem definition and the fairness notion we adopt, i.e. statistical (demographic) parity.
> The proposed loss functions can be viewed as a variation of Equation (21) in [Yazdani-Jahromi et al., NeurIPS 2024] (cited in our manuscript), and are explicitly grounded in demographic parity.
>
> In case of data with spurious correlations (like the Waterbird dataset), an approach based on demographic parity may lead to incorrect optimisation, not because of the loss function formulation, but rather because of a flaw in the fairness problem formulation. Let us explain our view in detail.
>
> Demographic parity assumes the presence of a minority group and is computed by comparing performance across two data partitions: one containing minority groups and one not containing minority groups.
>
> In conventional fairness benchmark datasets, a single sensitive attribute, such as gender, race, or age, can appropriately be used to generate disjoint partitions to isolate the minority group.
> For instance, in the ADULT dataset, one minority group is 'females earning more than $\$50k$'. This can be easily isolated with partitioning the data with the single attribute gender.
> Therefore, equalising the loss on the two partitions supports the fairness optimisation objective (regardless of and independently of the accuracy objective).
>
> However, when we attempt to apply the demographic-parity–based fairness approach to domains with more complex combinations of minority groups, precise partition definition is crucial and should be introduced.
>
> In particular, in the Waterbirds dataset, two minority groups, Waterbird on Land (WB-L) and Landbird on Water (LB-W), exist alongside two majority groups, Waterbird on Water (WB-W) and Landbird on Land (LB-L). If the background is used as the attribute for partitioning, both partitions encompass a minority group, leading to flawed demographic parity application to Fairness, where the learning process is skewed by majority group dominance in both partitions, as correctly noted by the reviewer.
>
> To correctly apply demographic parity for Fairness in the Waterbird dataset setting, the partitions should instead be defined as:
>  - Partition 1: WB-W and LB-L (the two majority groups),
>  - Partition 2: WB-L and LB-W (the two minority groups).
>
> As indeed suggested by the reviewer, this requires a combination of two dimensions, in this case, the class (bird type) and the 'sensitive' attribute (background).
>
> Under this construction, demographic-parity-based fairness encourages the model to treat images with birds in "natural" environments (majority groups) and "unnatural" environments (minority groups) equitably. In this way it is possible to train a model for spuriously correlated data like the Waterbirds adopting demographic parity approach.
>
> Our approach is tailored for conventional supervised fairness contexts with identified and labelled minority groups. Broadening our approach to encompass more complex configurations of minority groups requires an explicit and domain-specific "minority group" identification as suggested by the reviewer and demonstrated in techniques like GroupDRO.
>
> In our proposal, we intend to explicitly represent the Fairness problem separately from other considerations (e.g., main task performance). Of course, the appropriate identification of the minority groups is critical to the success of the application, but it is independent from the effectiveness of an MTL paradigm in which Fairness is induced in the internal representation and is not forced into the classification head.
>
> Regarding "advantage over existing approaches in terms of annotation requirements": in theory, one could consider extending Fairness to deal with spurious correlation automatically, i.e. without annotations. However, this seems to deviate from the scope and intentions of our work.
>
> Beyond clarifying the characteristics of the suggested loss functions, it is noteworthy to point out that the proposed methodology does not need to train the model from scratch, as demonstrated in the MTF experiment, where we introduce group-fairness by fine-tuning a pre-trained model.
>
> We sincerely thank the reviewer for his insightful comments. It is highly relevant and has helped us clarify the importance of defining the minority group/s. We will improve the definition of the method including a discussion on the relevance of identifying the appropriate partitions to support the demographic parity method for fairness. These additions will improve the clarity of our methodology and strengthen the overall presentation of the paper.

---

> ### Comment · Reviewer_7nuJ · 2025-11-25
>
> The reviewer sincerely thanks for the response from the authors. Indeed, the correct way to partition groups is the crucial problem, and I encourage the author to conduct the experiment in the future submission. As there are concerns for overfitting the minority group.
>
> Still, for the Waterbirds dataset, if we use the definition of the data partition based on your response (i.e., "Partition 1: WB-W and LB-L (the two majority groups), Partition 2: WB-L and LB-W (the two minority groups)".), the model could easily overfit the minority groups as the scarcity of data points. This may also lead to poor performance if we check the fairness metrics like EOp or EOdd. I encourage the author to further improve the method. Thank you once again for your effort on the rebuttal.

---

### Official Review · Reviewer_YzTD · 2025-10-31

**Soundness:** 2
**Presentation:** 2
**Contribution:** 2
**Rating:** 4
**Confidence:** 3

**Summary:**

This paper investigates fairness in machine learning by re-casting group-level fairness mitigation as an auxiliary task within a multi-task learning (MTL) framework. Rather than embedding fairness constraints solely as regularizers in the loss of a single task, the authors propose to decouple fair representation learning and prediction through concurrent, parallel optimization. The paper contributes two new fairness loss functions tailored for MTL auxiliary heads, provides generalization and fairness gap bounds connecting empirical and population objectives, and compares the proposed approach theoretically and empirically to regularization-based and adversarial methods across standard tabular and image benchmarks.

**Strengths:**

1. The central idea—framing fairness as an explicit auxiliary task in an MTL paradigm instead of as a loss regularizer—offers a clear perspective shift that addresses known trade-offs and optimization pitfalls in fairness-regularized learning.

2. Two new fairness-oriented loss functions (Group Loss Fairness [GLF] and Group Loss Divergence [GLD]) are constructed to operate independently of prediction heads; both are thoroughly defined (Equations 5–7, Pages 5–6) and well-motivated for MTL settings.

3. The experimental evaluation covers multiple standard tabular datasets (Adult Income, COMPAS, Bank Marketing) and one image dataset (MTF), using both binary and multi-group sensitive attributes. Results demonstrate the scalability and comparable or improved fairness outcomes of the proposed method versus established regularization and adversarial baselines.

**Weaknesses:**

1. **Missing Discussion of Directly Related MTL–Fairness Trade-off Papers**:
The literature review lacks a direct discussion and comparison with several closely related studies, particularly [1], which also addresses fairness through an MTL framework and appears to incorporate an additional fairness loss component (see Figure 4 in [1]). The authors should provide a more thorough discussion of how their method differs from or improves upon such prior work, ideally accompanied by empirical comparisons to clarify the relative advantages and limitations.

2. **Limited Explanation of Results Beyond Pareto and Aggregate Metrics**:
Although Figure 2 presents a favorable Pareto front, the accompanying discussion provides limited insight into the practical differences between the two proposed fairness losses—GLF and GLD. It remains unclear whether one loss consistently outperforms the other, or under what data conditions specific trade-offs emerge. Neither Table 1 nor the results narrative clarifies which component contributes most to the reported fairness gains. More detailed ablation studies or per-dataset analyses would strengthen the empirical evidence and help contextualize the observed improvements.

3. **Lack of Computational Cost Analysis**:
The paper omits a quantitative analysis of training efficiency. The added MTL heads and parallel objectives likely increase time, memory, and convergence cost, but these trade-offs are not reported.

4. **Unclear Presentation of Key Results**:
Some key results are difficult to interpret. For instance, the text and labels in Figure 2 and Table 1 are too small, making it hard to clearly compare different methods.

5. **Missing References to Advanced MTL Methods**:
The paper aims to address fairness using an MTL framework but lacks discussion of recent advanced MTL methods [1,2,3]. Including and positioning the proposed approach relative to these works would improve clarity and strengthen the paper’s contribution.

[1]. Understanding and Improving Fairness-Accuracy Trade-offs in Multi-Task Learning. KDD 2021

[2]. Fair resource allocation in multi-task learning. ICML 2024

[3]. Revisiting Fairness in Multitask Learning: A Performance-Driven Approach for Variance Reduction. CVPR 2025

**Questions:**

Refer to weaknesses.

---

> ### Author Response · Authors · 2025-11-21
>
> We respectfully clarify that our methodology aims to mitigate the fairness–accuracy trade-off using MTL, while prior work such as [1] addresses a different problem: ensuring fairness with respect to each tasks of an MTL model. Specifically, [1] formulates separate fairness objectives for each task using conventional fairness-regularised losses (e.g., FPR/TPR). Importantly, the fairness loss used in [1] is already included in our baselines, and because our experiments are conducted in a single-task setting, this provides a direct comparison with their proposed fairness regularisation component.
>
> We agree that a clearer discussion of the benefits and limitations of the different fairness loss functions considered would strengthen the paper. In a future version, we wish to expand the analysis to better articulate when and why each loss function performs differently.
>
> We wish to clarify that in the broader MTL literature, “fairness” has sometimes been used to describe the balance of performance amonge tasks, which is conceptually distinct from group-level fairness. For example, [2] studies gradient-based optimisation for resource allocation in wireless systems and applies gradient-surgery methods to assign in a "fair" way the resources (e.g. balancing the task performance). Similar ideas are explored in [3].
>
> Although such gradient-surgery techniques could potentially improve our MTL training, incorporating them would require an ablation study to unravel the source of performance gains. For clarity and focus, we therefore chose to evaluate our method without additional task-balancing optimisers such as [2] and [3]. In future extended version of this work we will consider to do that.
>
> We appreciate the discussion and the positive comments on our shift of perspective in addressing fairness task. The comments are useful to improve our work and we thank the reviewer for providing them.

---

> > ### Comment · Reviewer_YzTD · 2025-11-25
> > **Reply to authors**
> >
> > Thank you for your response. Although I share the other reviewers’ negative assessment of this submission, I still encourage the authors to further improve the paper by addressing the common concerns raised by the reviewers — including clarifying the methodological description, refining the experimental setup, and enhancing the overall presentation.

---

### Official Review · Reviewer_veg4 · 2025-11-03

**Soundness:** 2
**Presentation:** 2
**Contribution:** 1
**Rating:** 2
**Confidence:** 5

**Summary:**

The paper addresses group-level fairness in machine learning by reframing the problem as a multi-task learning (MTL) setting rather than embedding fairness directly as a regularizer in the predictive task. The authors propose modelling an auxiliary fairness task alongside the main predictive objective so that the network learns a latent representation that supports both accuracy and fairness. They introduce two novel fairness loss functions tailored for this MTL setup, provide a theoretical analysis of the generalization properties of their formulation, and empirically demonstrate on benchmark datasets that their MTL approach yields improved group-fairness metrics compared to both standard regularization-based fairness methods and other MTL variants, while maintaining competitive predictive performance.

**Strengths:**

The method is model-agnostic and can be integrated with existing architectures or loss functions without major redesign.

The paper offers generalization guarantees for the proposed MTL fairness formulation.

**Weaknesses:**

The paper’s core idea—disentangling task-related and fairness-related information—closely mirrors prior work [1], yet the draft does not cite or differentiate from it. It is highly recommended to add [1] to related work and make the distinction explicit (e.g., what is new in your objective, architecture, optimization, or theory), include ablations isolating your innovations, and provide a controlled head-to-head comparison under the same protocol and tuning budget to demonstrate non-trivial gains over [1].

The proposed method is only compared with very old benchmarks. The authors should incorporate more recent state-of-the-art baselines (contemporary MTL fairness methods, adversarial debiasing, group-DRO/thresholding, post-hoc calibration) and report fairness–utility frontiers to substantiate superiority.

Results are largely on toy datasets, leaving scalability and deployment unclear.

Multi-task heads add parameters and training time; no cost–benefit analysis or runtime/memory profile compared to simpler baselines is provided.

[1] Jang, Taeuk, and Xiaoqian Wang. "Fades: Fair disentanglement with sensitive relevance." Proceedings of the IEEE/CVF Conference on Computer Vision and Pattern Recognition. 2024.

**Questions:**

Please refer to the weaknesses.

**Details Of Ethics Concerns:**

I don't recognize significant ethical concerns.

---

> ### Author Response · Authors · 2025-11-21
>
> We thank the reviewer for the insightful comments and for pointing us to [1]. We agree that a clearer discussion of its relation to our work is important and will incorporate this in the future manuscript.
>
> However, our core objective is not to disentangle task-related and fairness-related components. Instead is to learn a fair representation.
> We use the auxiliary fairness task as a learning bias that encourages the shared latent representation to reduce demographic disparities while still supporting strong task performance. In our formulation, the model learns a single latent space that minimises group-dependent information within the shared representation, rather than explicitly disentangling the two aspects.
>
> This approach is important because, as also noted in [1] in section 2.1, approaches that attempt to enforce fair representations rely on restrictive assumptions and degrade performance—particularly in generative or vision settings where some sensitive attributes correlate with visual fidelity. Our multi-task learning design directly addresses this limitation by promoting fair internal representation without requiring disentanglement and without discarding information critical for the main task. We will make this conceptual difference explicit and add a dedicated comparison and discussion in the related work section.
>
> We also appreciate the suggestion regarding comparisons between fair-representation learning methods and disentanglement-based approaches. While such a broader empirical comparison would indeed be interesting, it lies outside the scope of the specific contribution we focus on in this work. Our goal here is to provide a clearer and more principled way to encourage fair internal representations through multi-task learning, rather than to disentangle fairness-related and task-related components.

---

### Meta-Review · Area_Chair_uy1E · 2026-01-08

**Summary:**

This paper frames fair representation learning as a multi-task learning (MTL) problem, treating fairness as an auxiliary task and proposing two fairness losses together with theoretical generalization analysis and experiments on standard tabular benchmarks. Across the discussion, reviewers generally agree the topic is relevant and the paper has a non-trivial amount of theory/experiments, but the rejection is driven by two persistent issues: (i) unclear novelty/distinction from closely related fairness/MTL formulations, and (ii) concerns that the proposed objective may not reliably capture commonly used fairness goals in realistic settings, especially when subgroup definitions are ambiguous or when there is risk of “overfitting” to minority groups. The rebuttal provides reasonable clarifications and promises additions (e.g., more related work, more discussion), and one reviewer indicates their initial concerns were largely addressed; however, the strongest skeptics remain unconvinced, and the overall empirical case still reads as not yet at the ICLR acceptance level.

**Reviewer Concerns:**

Novelty remains weak. Reviewer veg4 asks for a clearer statement of what is genuinely new  and a more controlled head-to-head comparison/ablation against strong alternatives, plus a clearer cost/benefit story. These are not fully closed by the discussion.

Fairness objective vs standard fairness metrics. Reviewer 7nuJ remains concerned that matching losses across sensitive groups does not necessarily translate to standard fairness goals and may even encourage undesirable behavior such as overfitting to minority groups, depending on the partition; the discussion does not fully resolve this.

Empirical evidence is still not strong enough. Reviewer YzTD highlights the missing discussion of directly related MTL–fairness trade-off lines of work and requests stronger, cleaner comparisons and improved presentation/experimental clarity. Reviewer 6fnZ also asks for a statistical analysis to justify the proposed loss term. After the rebuttal, these points are improved but not convincingly settled.

**Reviewer Scores:**

Reviewer veg4 (Rating 2: reject): likely stays at 2. The rebuttal does not fully deliver the controlled comparisons and sharper novelty/positioning they ask for.

Reviewer 7nuJ (Rating 2: reject): likely stays at 2. Their core objection about the fairness formulation/partition dependence and potential mismatch with EOp/EOdd remains.

Reviewer YzTD (Rating 4: marginally below threshold): likely stays around 4 (at best a weak 4). Their comment suggests alignment with the broader set of reviewer concerns even after the response.

Reviewer 6fnZ (Rating 4: marginally below threshold): could move slightly upward given they say the rebuttal addressed their initial concerns, but they also explicitly defer to the other reviewers’ remaining issues; overall this does not change the final recommendation.

---

### Decision · Program_Chairs · 2026-01-26

Reject